# Differences in Extracellular NAD^+^ and NMN Metabolism on the Surface of Vascular Endothelial Cells

**DOI:** 10.3390/biology11050675

**Published:** 2022-04-27

**Authors:** Patrycja Jablonska, Paulina Mierzejewska, Marta Tomczyk, Patrycja Koszalka, Marika Franczak, Ada Kawecka, Barbara Kutryb-Zajac, Alicja Braczko, Ryszard T. Smolenski, Ewa M. Slominska

**Affiliations:** 1Department of Biochemistry, Medical University of Gdansk, 80-211 Gdansk, Poland; patrycja.jablonska@gumed.edu.pl (P.J.); paulina.mierzejewska@gumed.edu.pl (P.M.); marta.tomczyk@gumed.edu.pl (M.T.); marika.franczak@gumed.edu.pl (M.F.); kaweckaada@gumed.edu.pl (A.K.); b.kutryb-zajac@gumed.edu.pl (B.K.-Z.); alicja.braczko@gumed.edu.pl (A.B.); ryszard.smolenski@gumed.edu.pl (R.T.S.); 2Institute of Medical Biotechnology and Experimental Oncology, Intercollegiate Faculty of Biotechnology of University of Gdansk and Medical University of Gdansk, Medical University of Gdansk, 80-211 Gdansk, Poland; patrycja.koszalka@gumed.edu.pl

**Keywords:** extracellular NAD^+^ metabolism, extracellular NMN metabolism, vascular endothelial cells, CD38, CD73

## Abstract

**Simple Summary:**

Nicotinamide adenine dinucleotide (NAD^+^) is a multifunctional metabolite involved in many key cellular processes. Outside the cell, NAD^+^ or its metabolites are important signaling molecules, related especially to calcium homeostasis, which controls the functioning of the heart. The cleavage of NAD^+^ or its precursor, nicotinamide mononucleotide (NMN), produces derivatives entering the cell to rebuild the intracellular NAD^+^ pool, which is important for cells with high energy turnover. Abnormalities in NAD^+^ and NMN metabolism can lead to cell aging and the development of cardiovascular diseases. In this study, we demonstrated that the extracellular metabolism of NAD^+^ and NMN is vastly different in the vascular endothelium obtained from different species and locations. This may have implications for strategies to modulate the NAD^+^ system and may cause difficulties for comparing the results of different reports.

**Abstract:**

The disruption of the metabolism of extracellular NAD^+^ and NMN may affect related signaling cascades and pathologies, such as cardiovascular or respiratory system diseases. We aimed to study NAD^+^ and NMN hydrolysis on surface endothelial cells of diverse origins and with genetically modified nucleotide catabolism pathways. We tested lung endothelial cells isolated from C57BL/6 J wild-type (WT) and C57BL/6 J CD73 knockout (CD73 KO) mice, the transfected porcine iliac artery endothelial cell line (PIEC) with the human E5NT gene for CD73 (PIEC CD73), and a mock-transfected control (PIEC MOCK), as well as HMEC-1 and H5V cells. Substrate conversion into the product was followed by high-performance liquid chromatography (HPLC). We showed profound differences in extracellular NAD^+^ and NMN metabolism related to the vessel origin, species diversity, and type of culture. We also confirmed the involvement of CD38 and CD73 in NAD^+^ and NMN cleavage.

## 1. Introduction

The primary metabolic function of nicotinamide adenine dinucleotide (NAD^+^) is related to redox reactions [1], but, in addition, NAD^+^ may also serve as a substrate in various signaling processes. The main enzyme classes responsible for the catabolism of NAD^+^ in the cell are sirtuins, poly(ADP-ribose) polymerases (PARPs), and cyclic ADP-ribose synthases/NAD^+^-glycohydrolases. Enzymes that participate in NAD^+^-dependent signaling pathways are involved in controlling the necessary balance of NAD^+^ concentration as well as cell cycle progression, transcriptional regulation, and DNA repair, and have, therefore, been identified as promising targets in many diseases [2,3,4].

Maintaining a constant level of NAD^+^ in a cell is essential for its proper functioning. Disturbed balance in the processes of NAD^+^ synthesis and degradation may be the source of the development of many pathologies. The main approach to regulating NAD^+^ levels in cells is through the efficient recycling of endogenous precursors. Nucleotide derivatives can support the maintenance of intracellular NAD^+^ pools by passing into the cell from the extracellular space [5]. Extracellular NAD^+^ is degraded to pyridine and purine metabolites by different types of surface-located enzymes, which are widely expressed on the plasma membrane of various cells and tissues [6]. The enzymes that take part in the degradation of extracellular NAD^+^ are mainly CD38-NAD^+^-glycohydrolase, which hydrolyzes NAD^+^ to nicotinamide (Nam) and ADP-ribose (ADPR) and nicotinamide mononucleotide (NMN) to Nam, and CD73-ecto-5′-nucleotidase, which is responsible for the hydrolysis of NAD^+^ to NMN and AMP, then NMN to nicotinamide riboside (NR). Enzymes of minor significance, but that should also be mentioned, are ecto-nucleotide pyrophosphatase/phosphodiesterase (eNPP1, CD203a) and alkaline phosphatase (ALP). All of these enzymes are known to exist on the surface of endothelial cells. However, the importance of extracellular NAD^+^ metabolism has not yet been thoroughly investigated [7,8].

The endothelium is a single layer of physiologically active cells that line the luminal surface of the entire vascular system [9]. It takes part in the secretion and active transport of chemical substances. The most important are prostacyclin (dilates blood vessels and inhibits platelet aggregation), nitric oxide (dilates vessels and inhibits platelet aggregation), von Willebrand factor (stimulates platelet aggregation), and thrombomodulin (inhibits blood clotting). The vascular endothelium takes part in vasoconstriction and vasodilation, and thus influences the regulation of blood pressure and tissue blood supply. It is involved in blood clotting (hemostasis and fibrinolysis), processes of atherosclerosis and angiogenesis, as well as inflammatory reactions. Endothelial dysfunction is a known hallmark of many vascular diseases [10,11]. Extracellular enzymes located on endothelial cells are an integral part of metabolism, which enable the homeostatic integration and control of the vascular inflammatory and immune responses of cells at the site of injury. The continuous development of therapeutic strategies targeting ectonucleotidases shows promise in the treatment of vascular thrombosis, impaired inflammation, and abnormal immune reactivity [12,13].

In this work, we measured the hydrolysis of extracellular NAD^+^ and NMN on different types of endothelial cells in terms of vascular origin, species diversity, and type of culture (immortalized or primary cell line). We have shown some significant differences that will help to understand the heterogeneity of NAD^+^-dependent metabolism and may constitute potential therapeutic strategies in the future.

## 2. Materials and Methods

### 2.1. Reagents

Adenosine (CAS 58-61-7), ADPR (adenosine 5′-diphosphoribose, CAS 68414-18-6), AOPCP (adenosine-5′-alpha,beta-methylene diphosphate, CAS 3768-14-7), deamino-NAD^+^ (nicotinamide hypoxanthine dinucleotide sodium salt, CAS 104809-38-3), nicotinamide (CAS 98-92-0), NAD^+^ (nicotinamide adenine dinucleotide, CAS 53-84-9), NMN (nicotinamide mononucleotide, CAS 1094-61-7), Dulbecco’s modified Eagle’s medium (DMEM), fetal bovine serum (FBS), Hanks Balanced Salt Solution (HBSS), L-glutamine, phosphate-buffered saline (PBS), penicillin/streptomycin, RPMI1640 medium, trypsin-EDTA solution, and endothelial cell growth supplement (ECGS) were obtained from Sigma-Aldrich (Poznan, Poland).

### 2.2. Animals Maintenance and Murine Lung Endothelial Cells Isolation

All experiments on mice were conducted following a Guide for the Care and Use of Laboratory Animals published by the European Parliament, Directive 2010/63/EU, and were performed with the approval of the Local Ethical Committee for Animal Experimentation in Bydgoszcz (27/2016). C57BL/6 J CD73 knockout (CD73 KO) mice were obtained from Heinrich-Heine-Universität in Düsseldorf, Germany [13]. The isolation of murine lung endothelial cells (LECs) from C57BL/6 J wild-type (WT) (*n* = 6) and CD73 KO (*n* = 5) mice was described previously [14]. Briefly, mice were fed a standard chow diet until 8 weeks and were then anesthetized with a mixture of ketamine/xylazine followed by opening the chest. Murine lungs were harvested, minced on a Petri dish, incubated with collagenase A (2.5 mg/mL solution in 0.1% BSA in HEPES), and then filtered through a 70-μm strainer. The cells were washed with PBS, resuspended in DMEM with D-valine, 10% FBS, ECGS (15 mg/500 mL), 2 mM L-glutamine, and 1% penicillin-streptomycin (*v*/*v*), and plated into a 25 cm^2^ tissue-culture flask. After reaching the specified density, cells were sorted using mouse CD31 MicroBeads on the MACS column (Miltenyi Biotec, 130-097-418) according to the manufacturer’s protocol. The sorted cells were further cultured in an endothelial cell medium. For the experiments, cells between two to five passages were plated at 12–16 × 10^4^ cells/well in 24-well plates. The cells were maintained in a 5% CO_2_ humidified atmosphere at 37 °C.

### 2.3. Cell Culture Conditions of Other Endothelial Cell Types

The endothelial cell lines human dermal microvascular endothelial cells (HMEC-1), murine immortalized heart endothelial cells (H5V), and transfected porcine iliac artery endothelial cells (PIEC) with human E5NT gene (PIEC CD73), as well as mock-transfected control (PIEC MOCK), were used in this study. The PIECs were kindly provided by Marco de Giorgi, who transfected them using F2A (the 2A sequence of Foot and Mouth Disease Virus) technology and Lipofectamine 2000 (Invitrogen, Waltham, MA, USA), and the transfection efficiency was evaluated by FACS analyses [8]. HMEC-1 were grown in MDCB 131 medium supplemented with 10% FBS, ECGS (15 mg/500 mL), hydrocortisone (1 µg/mL), 2 mM L-glutamine, and 1% penicillin-streptomycin (*v*/*v*). H5V cells were cultured in DMEM with 4.5 g/L glucose medium supplemented with 10% FBS, 2 mM L-glutamine, 1 mM sodium pyruvate, and 1% penicillin/streptomycin (*v*/*v*). PIEC MOCK and PIEC CD73 were cultured in RPMI 1640 medium supplemented with 10% FBS, 2 mM L-glutamine, and 1% penicillin–streptomycin (*v*/*v*). Cells were maintained in a 5% CO_2_ humidified atmosphere at 37 °C. Cells were cultivated in 25 cm^2^ and 75 cm^2^ tissue-culture flasks. For the experiments, the cells were plated at 4–6 × 10^4^ cells/well in 24-well plates. The medium was changed the next day, when the cells had adhered.

### 2.4. Determination of Cell-Surface NAD^+^ and NMN-Degrading Activities in Cell Cultures

After reaching 90–100% confluence in 24-well culture plates, LECs WT, LECs CD73 KO, HMEC-1, H5V, PIEC MOCK, and PIEC CD73 were rinsed with PBS, and 1 mL HBSS was added to each well. Then, to measure the total NAD^+^ and NMN hydrolysis activity, NAD^+^ or NMN (in a final concentration of 50 µM [15]) was added to the buffer and after 0, 30, 60, and 120 min of incubation at 37 °C, samples were collected and analyzed with the HPLC method using the LC system (Agilent Technologies 1100 series, Santa Clara, CA, USA) as described earlier [16]. The sample peaks were integrated and quantified using a ChemStation (Agilent Technologies, Santa Clara, CA, USA) chromatography data system (Appendix A).

The cell residue was dissolved in 0.5 mol/L NaOH, and the protein concentration was measured with the Bradford method according to the manufacturer’s protocol. The results of the cell-surface NAD^+^ and NMN degrading activities were expressed as the sum of products increased over time (nmol/min/g of protein).

### 2.5. Immunofluorescence Analysis

The distribution of CD73 and CD38 in H5V, HMEC-1, PIEC CD73, and PIEC-MOCK was detected by immunofluorescent staining in a 96-well optical bottom plate at a density 1 × 104 cells/well in a total volume of 200 μL cell culture medium. After 24 h, the cells were rinsed 3 times with PBS and fixed in ice-cold methanol for 5 min, and then rinsed with PBS. To reduce non-specific antibody binding, the cells were incubated with PAD solution (1% Bovine Serum Albumin and 10% normal goat serum solution in PBS). Thereafter, the cells were incubated with rabbit polyclonal anti-CD73 (1:100, Novus, Centennial, CO, USA) and mouse monoclonal anti-CD38 (1:100, Novus, Centennial, CO, USA) primary antibodies diluted in PBS for 1 h at room temperature. Next, the cells were incubated with goat anti-rabbit antibodies labeled with Alexa Fluor 488 (1:600, JacksonImmuno, West Grove, PA, USA) and goat anti-mouse antibodies labeled with goat anti-mouse Alexa Fluor 594 (1:600, JacksonImmuno, West Grove, PA, USA) for 30 min at room temperature. Negative controls were obtained by incubation without primary antibodies (data not shown). The cell nuclei were stained with 4′,6-diamidino-2-phenylindole (DAPI) (1:1000, Thermo Fisher Scientific, Waltham, MA, USA) for 5 min. The stained cells were imaged and analyzed as described previously [17] with an AxioCam MRc5 camera and an AxioObserved.D1 inverted fluorescent microscope (Carl Zeiss AG, Jena, Germany), and analyzed using Zen image processing software (version 3.3). The total CD73 and CD38-positive areas in cells were measured and the mean fluorescence intensity was calculated.

### 2.6. Determination of Particular Ecto-Enzymes Engaged in the Extracellular NAD^+^ and NMN Catabolism on the Surface of the Endothelial Cells

To determine the specific cell-surface ecto-enzymes engaged in the metabolism of NAD^+^ and NMN, 1 mL of HBSS with inhibitors for CD38 and CD73 was added: a competitive inhibitor of CD38 (150 µM deamino-NAD^+^, Appendix A) and CD73 inhibitor (50 µM AOPCP [18]), respectively. NAD^+^ and NMN were added as substrates at a final concentration of 50 µM, respectively. After 0, 30, 60, and 120 min, 50 µL of the sample was collected, centrifuged (14,000× *g*/15 min/4 °C), and analyzed by HPLC. The degradation rates were shown as nmol/min/g of protein.

### 2.7. Statistical Analysis

Values were presented as the mean ± SEM. Statistical analysis was performed using an unpaired Student’s *t*-test and one- or two-way ANOVA followed by Tukey’s or Bonferroni’s posttest (GraphPad software, San Diego, CA, USA). A *p*-value of 0.05 was considered to indicate a significant difference.

## 3. Results

### 3.1. NAD^+^ and NMN Hydrolysis Are Different for Various Endothelial Cell Types

The comparison of the total NAD^+^ and NMN hydrolysis on endothelial cell lines showed that the metabolism differed depending on the species and organ origin. The most active cells were transfected PIEC with a gene for CD73 (Figure 1a,b). The PIEC CD73 had the strongest signal for CD73 in immunofluorescent staining (Figure 1e), confirming the successful transfection of the NT5E gene compared to the PIEC MOCK control (Figure 1f). H5V, on the other hand, had the lowest signal for CD73 (Figure 1d), which translated into lower total NMN hydrolysis (Figure 1b). CD38 was also present in the cell types tested; however, despite its presence on HMEC-1 cells, no NAD^+^ hydrolysis was observed in the analyzed incubation time (Figure 1a).

In contrast to NAD^+^ hydrolysis, total NMN hydrolysis was observed on the surface of HMEC-1 after 2 h of incubation (Figure 1b). This may be due to the significant presence of CD73 on these cells (Figure 1g). Small amounts of NMN were also metabolized on H5V and PIEC MOCK cells. Figure 1c shows a diagram of NAD^+^ and NMN degradation in the extracellular space. CD73 is an abundant endothelial enzyme responsible for the hydrolysis of NAD^+^ to NMN and AMP, and then to NR and Ado, respectively. CD38, in turn, is involved in the metabolism of NAD^+^ and NMN, leading to the production of Nam.

### 3.2. PIEC Cells Mainly Produce NMN, AMP, and NR

In NAD^+^ hydrolysis on the surface of PIEC MOCK cells (Figure 2a), the main products are NMN and AMP. In the case of the transfected PIEC CD73, the hydrolysis was enhanced and was characterized by a more intensive transformation of NAD^+^. In addition to NMN and AMP, and other products of their metabolism (such as NR, Ado, and Ino), Nam was also created. This indirectly confirms the successful transfection of NT5E, the gene for CD73 that is responsible for the hydrolysis of NAD^+^ to NMN and NMN to NR, but also leads to the generation of AMP from NAD^+^ and further Ado. For NMN hydrolysis, the only product was NR (Figure 2b). When comparing PIEC MOCK and PIEC CD73 cells, there was about a 10-fold increase in the concentration of the nascent NR.

### 3.3. Nam and ADPR Are the Main Products of NAD^+^ and NMN Metabolism on the Surface of LECs

LEC cells were the most metabolically active in NAD^+^ and NMN hydrolysis (Figure 3a,b). The main products that were formed on the surface of the endothelium were Nam and ADPR, followed by Ado in the case of NAD^+^ hydrolysis (Figure 3c), and Nam and NR in the case of NMN hydrolysis (Figure 3d). In NAD^+^ hydrolysis, cells isolated from CD73 KO mice showed lower conversion to Ado and reduced Nam formation (Figure 3c). In NMN hydrolysis, less NR was formed on LECs CD73 KO cells compared to the control (Figure 2b). This could confirm the involvement of CD73 in the formation of these products on the surface of the lung endothelium.

### 3.4. LEC and PIECs Are Characterized by Different NAD^+^ and NMN Metabolism

When we compared the substrate consumption of NAD^+^ and NMN on LECs cells, as well as PIEC, in an experiment with inhibitors of the main enzymes involved in the metabolism of extracellular NAD^+^ and its derivative NMN, we observed that, after the inhibition of CD38 activity on the surface of LECs, the NAD^+^ concentration significantly increased on both LECs WT and LECs CD73 KO (Figure 4a). The knockout of CD73 caused less NAD^+^ consumption, suggesting that CD73 may also be involved in its metabolism. Due to the lack of sufficient cells for the experiment, we did not perform CD38 inhibition for NMN hydrolysis. Nevertheless, we can assume that the formation of Nam and NR that we observed in Figure 2b is due to both CD38 and CD73 activity, since CD73 knockout affects NMN consumption (Figure 4b).

A characteristic feature of PIEC CD73 cells is the increased metabolic activity of both NAD^+^ and NMN (Figure 4c,d) due to the insertion of the additional human CD73 gene. The inhibition of CD73 (by AOPCP) did not affect the NAD^+^ hydrolytic activity on the surface of PIEC MOCK cells, while it significantly reduced NAD consumption in PIEC CD73 cells. Moreover, the inhibition of CD38 by dNAD had no such effect (Figure 4c).

The NMN intake on PIEC CD73 cells was mainly due to the activity of CD73, as shown in Figure 4d. The inhibition of CD38 did not reduce NMN consumption on the PIEC CD73; the cells behaved as though they were untreated, suggesting that this enzyme was only slightly involved in NMN metabolism. In contrast, on PIEC MOCK, no significant changes in NMN consumption were observed following the inhibition of CD38 and CD73.

## 4. Discussion

In this study, we compared the metabolism of NAD^+^ and NMN in various vascular endothelial cells and confirmed that CD38 and CD73 are the main enzymes responsible for their hydrolysis. In addition, our studies showed that endothelial cells are characterized by different metabolisms of NAD^+^ and NMN. In the case of NAD^+^ metabolism on HMEC-1 cells, we demonstrated its absence within the studied time frame as compared to NMN, which may indicate a diverse affinity of the enzyme for the substrate. Additionally, differences in the metabolism of NAD^+^ and NMN were observed due to the type of cell culture conducted (primary or immortalized cells) and due to their species and tissue origin.

Extracellular NAD^+^ is a known signaling molecule that can act directly or indirectly through its metabolites at purinergic receptors and modulates many functions, including inflammatory processes and calcium signaling [19,20]. NAD^+^ appears in the extracellular space through lytic and non-lytic mechanisms, such as connexin 43 [21]. The major metabolites of NAD^+^ conversion are NMN and AMP (CD38, CD73, and CD203a activity), Nam and ADPR (CD38 activity), and NR, as a result of further metabolism of NMN (CD73 activity) [22]. In our research, we confirmed the presence of these pyridine derivatives. Furthermore, AMP was cleaved to Ado (CD73 activity) and further to Ino in PIEC CD73 cells. This is consistent with what we observed in endothelial cells when investigating the extracellular metabolism of adenine nucleotides [17]. In LECs, there were Nam and ADPR, and then Ado production. NMN, Nam, and NR metabolites could restore the intracellular NAD^+^ pool by the NAD^+^ salvage pathway [23]. Furthermore, studies about NAD^+^ metabolism and its precursors have great value [24]. There is ample evidence that the administration of NAD^+^ precursors has systemic benefits. For example, administered NR restores the tissue levels of NAD^+^ and increases autophagy, possibly serving as a protective response in acute kidney injury (AKI) [25]. NR supplementation and CD38 inhibition directly suppress neuroinflammation in the brain by boosting NAD^+^ [26]. NMN supplementation, likewise, prevents age-associated gene expression changes in essential metabolic organs and improves mitochondrial oxidative metabolism in skeletal muscle in mice [27]. Nowadays, the importance of NMN has increased and it has been assigned the function of a signaling molecule that interacts between three organs: the hypothalamus, adipose tissue, and skeletal muscle [28,29]. Moreover, interest in this compound is growing since the plasma transporter for NMN has been recently discovered [30], although not all approve of this finding [31]. Research into the beneficial effects of NAD^+^ enhancement therapies has also been conducted in humans [32]. In the case of administration to Nam, it is less important due to the controlled process of resynthesis to NAD^+^ and shorter retainment in the body than NMN [33]. Moreover, Nam may cause hepatotoxicity or flushing in high-dose medications [34].

This study showed that primary cells isolated from the lungs of WT and CD73 KO mice presented active hydrolysis of NAD^+^ to Nam and ADPR by CD38 on their surface. Moreover, LECs were characterized by the highest degree of hydrolysis of both NAD^+^ and NMN among the tested cell lines. Regardless of the lower activity of CD73, CD38 is the predominant enzyme on the surface of the lung endothelium. This is consistent with the function of CD38 in the respiratory system. NAD^+^ metabolites formed by CD38 (ADPR and adenosine) play a role in intracellular calcium regulation in various cell types, including airway smooth muscle (ASM) cells [35]. They contribute to airway inflammation and hyperresponsiveness [36,37]. There are also reports that CD38 knockout suppresses tumorigenesis in mice and the clonogenic growth of human lung cancer cells [38,39]. In our study, we also proved high CD38 activity on primary lung endothelial cells.

On the other hand, studies on PIEC cells showed low activity of CD38 in these cells and confirmed that CD73 is the main enzyme responsible for the hydrolysis of NAD^+^ and NMN. The transfection of PIEC cells by the human NT5E gene increased the degrading activity of NAD^+^ and NMN by about three times. The extracellular conversion of NMN to NR by CD73 localized in the luminal surface of endothelial cells represents important vasoprotective mechanisms maintaining intracellular NAD^+^ and the healthy phenotype of endothelial cells [35]. Much research on CD73 activity has been aimed at exploring adenosine-dependent mechanisms (AMP to adenosine hydrolysis). In our experiments, we have also tried to show the important role of the transformation of NAD^+^ and NMN to NR by CD73.

An interesting result was the observation of the lack of NAD^+^-cleavage activity during the two-hour incubation and the remark of NMN metabolism under the same conditions. This may indicate different affinities of enzymes for NAD^+^ and NMN substrates. Differences in the Michaelis constant and Vmax were observed in the work by Mateuszuk et al. [40]. They compared the metabolism of extracellular NMN by CD73 and CD38 on Eahy.926 cells. Moreover, they showed a greater contribution of CD73 to NMN conversion in human endothelial cells compared to CD38, previously identified as the major NMN-degrading enzyme in mouse tissues in vivo [41].

Despite the importance of NAD^+^ metabolism to human health and diseases, determining the levels of NAD^+^ remains a challenge. Our research may also have some limitations. The reagents used for cell culturing were aimed at maximizing the growth and survival of cells in cultures and did not fully reflect the biological processes taking place in the body in vivo. FBS is an example of a compound where the presence of nucleotide pyrophosphatase and 5-nucleotidase activities has been discovered [42]. In our research, we also used this reagent. Each of the tested lines had the same conditions, so the impact on the experiment was similar. Another limitation in comparing the results with other studies may be the way of measuring NAD^+^ and its metabolism. In our research, we relied on the measurement of the concentrations of the resulting products using reversed-phase HPLC [16]. Other authors are used to NMN spectroscopy [43]; however, the gold standard is LCMS [44,45].

To summarize, NAD^+^ and NMN metabolism has been the subject of many studies, including brain cells [46], stem cells [47], muscles [48], fibroblasts [49,50], inflammatory cells [51], and WAT and BAT [52]. Our group has also examined NAD^+^ and NMN hydrolysis on the surface of human aortic valves and vessels [53]. Endothelial cells are a common model in the study of inflammation, circulation, leukocyte transport, angiogenesis, and cancer research [54,55,56]. Recently, many associations with endothelial dysfunction and NAD^+^ deficits, as well as CD38 activity, in the development of COVID-19 have also been demonstrated [57,58,59,60]. The differences observed in this study emphasize that the extracellular metabolism of NAD^+^ and NMN are heterogeneous and results may vary depending on the tested material. This is further evidence that research on changes in the extracellular NAD^+^ levels has its limitations and there is a lack of consistent standardization.

## 5. Conclusions

The extracellular metabolism of NAD^+^ and NMN is active on endothelial cells, but the exact rates differ depending on the cell type and conditions. Primary cultures are characterized by the highest degree of NAD^+^ and NMN hydrolysis compared to cell lines. Moreover, the presence of enzymes varies between species. We have demonstrated the presence of CD73, which was present in abundance on human HMEC-1 cells, while the murine cells (H5V) had less of this enzyme’s activity. In the case of CD38, the amount of this enzyme on cell lines was similar, but its activity was highest in primary cells isolated from mice lungs. This may be related to the role CD38 plays in the respiratory system. On the other hand, in experiments using cells with CD73 overexpression, a significant increase in the metabolism of NAD^+^ and NMN was observed, leading to the formation of Ado, which can be used, for example, in antiarrhythmic therapy. Due to differences in NAD^+^ and NMN metabolism, as well as in the way this metabolism is measured, the results obtained with cell culture experiments should be compared with caution.

## Figures and Tables

**Figure 1 biology-11-00675-f001:**
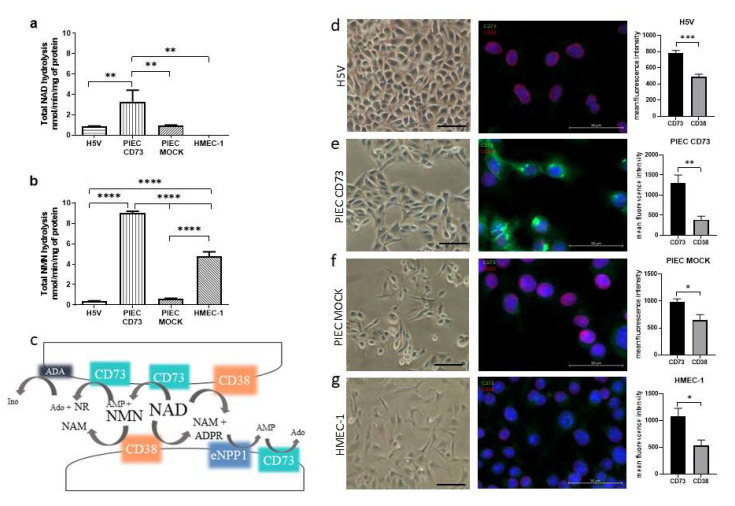
Nicotinamide adenine dinucleotide (NAD^+^) and nicotinamide mononucleotide (NMN) hydrolysis are different for various endothelial cell types. Comparison of total NAD^+^ (**a**) and NMN (**b**) hydrolysis on endothelial cell lines. Results are presented as the mean ± SEM; *n* = 3–6; ** *p* < 0.01 **** *p* < 0.0001 with one-way ANOVA followed by Tukey’s multiple comparisons test. Diagram of NAD^+^ and NMN metabolism in the extracellular space (**c**). Immunofluorescence staining and quantification for CD73 (green signal) and CD38 (red signal) on cell lines. Cell nuclei were counterstained with DAPI (blue signal) for H5V (**d**), PIEC CD73 (**e**), PIEC MOCK (**f**), and HMEC-1 (**g**). Results are presented as the mean ± SEM; *n* = 5; * *p* < 0.05, ** *p* < 0.01, *** *p* < 0.001 by student unpaired *t*-test.

**Figure 2 biology-11-00675-f002:**
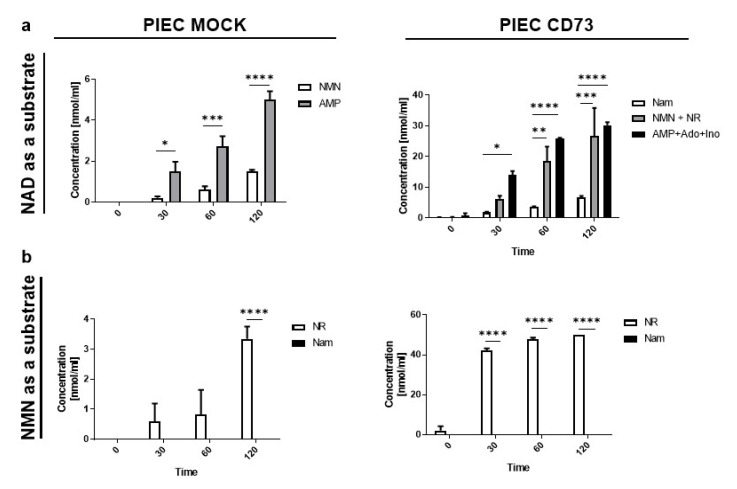
Transfected pig endothelial cells (PIEC) with the human CD73 gene show enhanced NAD^+^ and NMN metabolism. Concentration of products after incubation with 50 μM NAD^+^ (**a**) and 50 μM NMN (**b**) on the PIEC MOCK (control) and PIEC CD73. Results are presented as the mean ± SEM; *n* = 3–6 with two-way ANOVA followed by Tukey’s posttest. * *p* < 0.05, ** *p* < 0.01, *** *p* < 0.001, **** *p* < 0.0001.

**Figure 3 biology-11-00675-f003:**
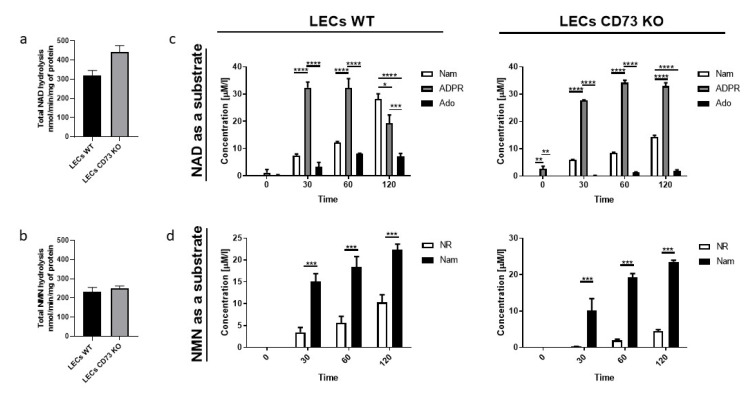
NAD^+^ and NMN metabolism are different on LECs isolated from WT mice and CD73 KO mice. Total NAD (**a**) and NMN (**b**) hydrolysis on LECs WT and LECs CD73 KO. Concentration of products after incubation with 50 μM NAD^+^ (**c**) and 50 μM NMN (**d**) on the LECs WT and LECs CD73 KO. Results are presented as the mean ± SEM; *n* = 3–6. * *p* < 0.05, ** *p* < 0.01, *** *p* < 0.001, **** *p* < 0.0001 with two-way ANOVA followed by Bonferroni’s posttest.

**Figure 4 biology-11-00675-f004:**
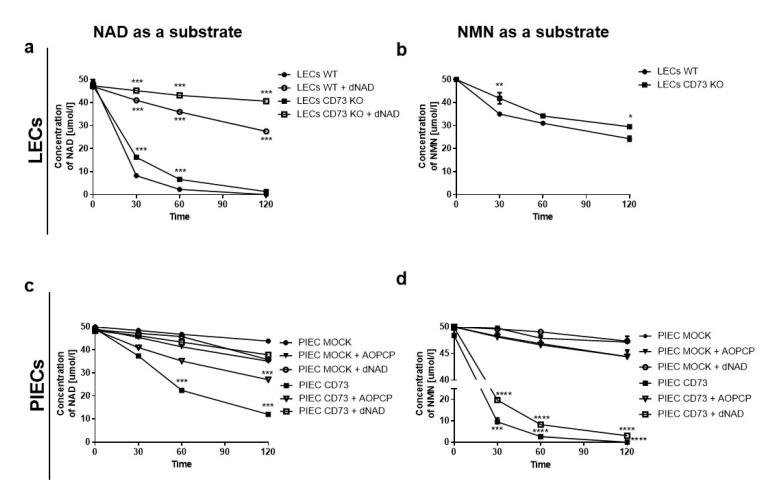
LEC and PIECs characterized by different NAD^+^ and NMN consumption—CD38 is mainly present on the LECs, while CD73 is primarily characteristic of PIEC. Mean comparison of the changes in the NAD^+^ (**a**,**c**) and NMN (**b**,**d**) concentration over time on LECs WT and CD73 KO with an inhibitor of CD38–dNAD^+^ and CD73–AOPCP, and on PIEC MOCK and PIEC CD73 with these inhibitors. *n* = 3–6; * *p* < 0.05, ** *p* < 0.001, *** *p* < 0.001, **** *p* < 0.0001 vs. LECs WT or PIEC MOCK with a two-way ANOVA followed by Bonferroni’s posttest.

## Data Availability

The authors declare that the data supporting the findings of the study are available within the article.

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
