# Peer review of "Differences in Extracellular NAD^+^ and NMN Metabolism on the Surface of Vascular Endothelial Cells"

_biology, 2022, doi:10.3390/biology11050675_

Round 1
Reviewer 1 Report
It is a nice original paper by Jablonska et al. that demonstrated the metabolism of NAD+ and NMN on various vascular endothelial cells including lung endothelial cells isolated from 28 C57BL/6 J wild type (WT) and C57BL/6 J CD73 knockout (CD73 KO) mice, the transfected porcine iliac artery endothelial cell line (PIEC) with the human E5NT gene for CD73 (PIEC CD73), and a mock-transfected control (PIEC MOCK) as well as HMEC-1 and H5V cells and confirmed that CD38 and CD73 are the main enzymes responsible for their hydrolysis.
I have some suggestions below:
- The authors add NAD+ or NMN (in a final concentration of 50 μM) to measure total NAD+ and NMN hydrolysis activity. Please mention a reference for this concentration and explain how to choose this concentration.
- The authors added inhibitors of CD38 (150 μM deamino-NAD+) and CD73 inhibitor (50 μM AOPCP) to measure the specific cell-surface ecto-enzymes engaged in the metabolism of 140 NAD+ and NMN. Please mention reference for 50 μM AOPCP and 150 μM deamino-NAD+.
- The authors used transfected pig endothelial cells (PIEC) with the human CD73 gene. What method used for transfection? Have the authors assessed the efficiency of transfection?
- I suggest that the authors perform western blot for CD73 protein expression in transfected PIEC with the human CD73 gene.
- The authors can do a western blot for NAD and NMN.
- The authors can present some pictures as a phase contrast for the endothelial cells.
Reviewer 2 Report
Differences in extracellular NAD+ and NMN metabolism on the surface of vascular endothelial cells
In this manuscript, Jablonska et al. demonstrated variations in extracellular NAD+ and NMN hydrolysis on the surface of several vascular endothelial cells. Furthermore, they confirmed the involvement of enzymes such as CD38 and CD73 in NAD+ and NMN cleavage. Declining levels of NAD+ and NMN causes aging and various cardiovascular diseases. The overall manuscript is interesting, however, there are few things the authors need to work on.
- Could the authors mention what method was used to transfect CD73 gene to PIEC cells and provide some images?
- Could the authors provide references for the concentrations of NAD+ and NMN used for hydrolysis activity?
- The introduction section is well-written but there are some shortcomings in the results section. Please try to explain the results in detail including what may be the reasons behind these observations instead of just summarizing the trend in the graphs.
- Is there any difference between Nam and NAM? If it is the same, please be consistent throughout the paper.
- Please perform the statistical analysis for the graphs in Figure 3.
- AMP was cleaved to Ado and further to Ino in PIEC CD73 cells. Can the authors confirm the results using other assays or experiments?
Reviewer 3 Report
In this work, Jablonska et al examine the conversion of NAD and NMN by the enzyme CD73 on two different endothelial cells. For this purpose CD73 KO and also the overexpression of CD73 are used. The quantification is done by the method of HPLC.
I have major and minor revision points to this work.
Major:
1. you describe the hydrolytic effect of NAD+ and NMN by using different endothelial cells. You focus on CD73 KO and overexpression. In the discussion you mention in the first sentence that this is a confirmation that CD38 and CD73 are the main responsible enzymes. Please also give the amounts of the enzymes in relation to the different cells. For this, a western blot would already be sufficient.
2. please explain in more detail why CD38 and CD73 should be the main enzymes. For this statement, a CD38/CD73 KO (or knockdown) cell should also be characterized. Is there already literature on this or would an experiment using siRNA be a possibility?
3. please keep your data constant between the WT and KO or overexpression in the cells. Why are different analyses performed in Fig. 3 a between cells. You need to explain this in detail at least. The same is true for Fig. 4.
4. you discuss due to different amounts of metabolites between the MOCK and the overexpression but also between the WT and the KO cells. I understand that you use scale to do this. However, I strongly suggest at least considering an alternative representation. Maybe compare the measurements not by time but by experimental groups and thus show the kinetics in multiple graphs.
Minor:
1. please include a section on statistics.
2. fig. 2 has an error in the legend in part B.
3. please add a supporting figure showing the metabolic degradation of NAD+ and NMN. This would help better understanding and make the discussion easier.
4. please expand the description of the HPLC method. What machine did you use. Are there example images for the measurements (in suplement if necessary).
Round 2
Reviewer 1 Report
The manuscript has been improved, I have no further comments.
Reviewer 2 Report
The paper is well revised.
Reviewer 3 Report
I have no further comments. Thank you very much.